# Influences of Sex, Education, and Country of Birth on Clinical Presentations and Overall Outcomes of Interdisciplinary Pain Rehabilitation in Chronic Pain Patients: A Cohort Study from the Swedish Quality Registry for Pain Rehabilitation (SQRP)

**DOI:** 10.3390/jcm9082374

**Published:** 2020-07-25

**Authors:** Björn Gerdle, Katja Boersma, Pernilla Åsenlöf, Britt-Marie Stålnacke, Britt Larsson, Åsa Ringqvist

**Affiliations:** 1Pain and Rehabilitation Centre, Department of Health, Medicine and Caring Sciences, Linköping University, SE-581 83 Linköping, Sweden; britt.larsson@liu.se; 2School of Law, Psychology and Social Work, Örebro University, SE-701 82 Örebro, Sweden; Katja.Boersma@oru.se; 3Department of Neuroscience, Uppsala University, SE-751 05 Uppsala, Sweden; pernilla.asenlof@neuro.uu.se; 4Department of Community Medicine and Rehabilitation, Rehabilitation Medicine, Umeå University, SE-901 87 Umeå, Sweden; britt-marie.stalnacke@umu.se; 5Department of Neurosurgery and Pain Rehabilitation, Skåne University Hospital, SE-221 85 Lund, Sweden; Asa.Ringqvist@skane.se

**Keywords:** chronic pain, country of birth, education, sex, rehabilitation, interdisciplinary, outcome

## Abstract

This study investigates the effects of sex, education, and country of birth on clinical presentations and outcomes of interdisciplinary multimodal pain rehabilitation programs (IMMRPs). A multivariate improvement score (MIS) and two retrospective estimations of changes in pain and ability to handle life situations were used as the three overall outcomes of IMMRPs. The study population consisted of chronic pain patients within specialist care in the Swedish Quality Registry for Pain Rehabilitation (SQRP) between 2008 and 2016 at baseline (n = 39,916), and for the subset participating in IMMRPs (n = 14,666). A cluster analysis based on sex, education, and country of origin revealed significant differences in the following aspects: best baseline clinical situation was for European women with university educations and the worst baseline clinical situation was for all patients born outside Europe of both sexes and different educations (i.e., moderate-large effect sizes). In addition, European women with university educations also had the most favorable overall outcomes in response to IMMRPs (small effect sizes). These results raise important questions concerning fairness and equality and need to be considered when optimizing assessments and content and delivery of IMMRPs for patients with chronic pain.

## 1. Introduction

Approximately 20% of the European population has a significant chronic pain condition [1]. Persistent/chronic pain is influenced by and interacts with physical, psychological, social, and contextual factors [2,3,4]. Moreover, since chronic pain conditions are associated with increased psychological distress, poor health, sick leave, and high socioeconomic costs [5], a biopsychosocial approach is the foundation of modern clinical pain care and research [6].

Interdisciplinary multimodal pain rehabilitation programs (IMMRPs) are a subgroup of interdisciplinary treatments according to the International Association for the Study of Pain (IASP). The core goals of these programs are broad and multifactorial and are combined with the individual goals of the patient [7,8]. IMMRPs are well-coordinated and administrated over several weeks to few months in group settings by different health professionals [9,10,11,12]. IMMRPs, including those in Sweden, are most often psychologically-based interventions (generally, cognitive behavioral therapy (CBT) and often third wave CBT, i.e., acceptance and commitment therapy (ACT)). These treatments include group activities such as chronic pain education, supervised physical activity, and activity training coordinated by an interdisciplinary team [9,10,11,12].

Systematic reviews (SRs) report higher efficacy on a general level for specific outcomes of IMMRPs and with respect to reduced costs compared with single treatment or treatment-as-usual programs [10,12,13,14,15,16,17,18]. However, striving to detect causal relationships, SRs, and randomized controlled trials (RCTs) may suffer from various degrees of bias [19,20,21]. Thus, it is necessary to investigate whether the evidence obtained from SRs and RCTs can be replicated within a consecutive non-selected flow of patients in clinical settings. Few studies have investigated the real-life outcomes of IMMRPs for chronic pain, but a large study from the Swedish Quality Registry for Pain Rehabilitation (SQRP) found small to moderate effect sizes for 22 outcomes [22]. Complex interventions such as IMMRPs should have several outcomes measured at multiple levels and include strategies for handling multiple outcomes [7,8,23,24], but most SRs of IMMRPs evaluate the outcomes as independent from each other. However, changes in a substantial number of outcomes from IMMRPs are likely to be intercorrelated. This assumption was confirmed for the 22 outcome variables in the large SQRP study [22]. Based on these results, a multivariate improvement score (MIS) of the majority of these 22 variables was defined as an overall outcome measure of IMMRPs. The SQRP also contained two overall retrospective outcome measures concerning change in life situation and change in pain after the IMMRP [22]. These two retrospective variables supplement the MIS (which is based on 22 variables answered on up to three occasions), giving a more complete picture of the outcomes. In our recent study, the majority of patients reported partially/markedly improved situations both immediately after IMMRP and at the 12-month follow-up [22].

On the request of the Swedish government in 2019, the Swedish Agency for Health Technology Assessment and Assessment of Social Services (SBU) reviewed the literature on treatments for chronic pain from a sex perspective. This effort was motivated by the recognition that women had a higher prevalence of chronic pain in the context of Swedish health care services and work compensation. The report concluded that few SRs specifically investigated whether the effects of IMMRPs are moderated by sex [25]. The question that arises and seems clinically important to address is whether women and men in the non-selected flow of patients in clinical settings have different IMMRP outcomes. Between 2008 and 2016, women compared to the community prevalence were overrepresented in Swedish specialist clinics. This overrepresentation is further increased with respect to participation in IMMRPs [26,27]. There are other demographic factors aside from sex that are important to consider. Patients with chronic pain included in the SQRP had a lower mean education than the population [28]. The impact of pain and disability is strongest among socially disadvantaged populations and among those with low education [29,30]. This may indicate socio-economic influences on IMMRP outcomes. In addition, migration is a critical factor of health inequalities [31,32]. Associations between migration and mental health as well as chronic pain have been reported [33,34,35,36,37,38,39]. Hence, according to some reports, immigrants experienced a greater impact of chronic pain (e.g., pain prevalence and higher pain intensity) than natural born inhabitants [38,39]. Currently, there is a lack of consistency if the IMMRP results for immigrants are equivalent to those for natives [38,40,41,42,43,44].

An increased knowledge about whether and how sex, education, and country of birth are associated with clinical presentations and outcomes of IMMRPs may influence future methods of clinical assessments and designs of IMMRPs. The need for more knowledge about these factors motivated this study.

We hypothesized that the clinical presentations and overall outcomes of IMMRPs in specialty care are influenced by sex, education, country of birth, and the combination of these factors. Hence, this study investigates whether these factors were associated with the clinical presentations at baseline and with the overall outcomes of IMMRPs immediately after the treatment and at a 12-month follow-up. Three global variables were used as main overall outcomes: the MIS and two retrospectively designed items concerning change in life situation and pain after IMMRPs. To deepen the clinical understanding of the MIS results, we also included results for the 22 variables used to obtain this global variable.

## 2. Experimental Section

### 2.1. Materials and Methods

As the SQRP and the instruments included have recently been described in detail elsewhere, they are only briefly described below [22].

#### 2.1.1. The Swedish Quality Registry for Pain Rehabilitation (SQRP)

The SQRP receives data from all relevant specialist clinics in Sweden [45]. The SQRP is based on patient reported outcome measures (PROM) questionnaires and captures a patient’s background, pain intensity, pain-related cognitions, and psychological distress symptoms, as well as activity/participation aspects and health-related quality of life variables. Patients complete the questionnaires on up to three occasions: (1) before assessment on the first visit to the clinic (baseline or pre IMMRP); (2) immediately after IMMRPs (post IMMRPs); and (3) at the 12-month follow-up after IMMRP discharge (12-month follow-up). Not all patients assessed will participate in IMMRPs. Some may need further investigation, some may need to undergo unimodal treatments, or some may, for different reasons, not participate in the IMMRPs despite recommendation at the assessment.

#### 2.1.2. Subjects

This study included SQRP data from subjects ≥18 years old with complex chronic (≥3 months) non-malignant pain who were referred to specialist care centers between 2008 and 2016. Residents in Sweden, including those with permanent residence permits, can seek health care for both acute and chronic conditions, while visitors from other countries can receive emergency medical care. Patients with pain first seek care/rehabilitation in primary care and if the responsible physician concludes that the chronic pain condition cannot be handled satisfactorily at this level of care, the patient is referred to the specialist care. The exact proportion of patients within primary health care with chronic pain conditions is unknown, but reasonable estimations are 10–20% [46,47]. The exact proportion of patients with chronic pain within primary health care who are referred to specialist clinics is also unknown. One can assume that patients referred to specialist care did not respond to routine pharmacological and/or physiotherapeutic treatments delivered in a monodisciplinary fashion. Because they often have comorbidities such as depression, anxiety, and kinesiophobia, their condition can be characterized as complex with higher severity of pain [26,48]. Strict criteria for inclusion in the registry are not available. However, general inclusion criteria for IMMRPs were: (i) disabling non-malignant chronic pain (on sick leave or experiencing major interference in daily life due to chronic pain); (ii) age 18 years and above; (iii) no further medical investigations needed; and (iv) written consent to participate and attend IMMRP. General exclusion criteria for IMMRPs were severe psychiatric comorbidity, abuse of alcohol and/or drugs, diseases that did not allow physical exercise, and specific pain conditions with other treatment options available (for instance saddle anesthesia or history of carcinoma). In the initial clinical assessment, the patient not fluent in Swedish can receive assistance with an interpreter. Patients who participate in IMMRP—including answering the questionnaires of SQRP—must be able to access the content and be able to communicate in Swedish.

The study was conducted in accordance with the Helsinki Declaration and Good Clinical Practice and approved by the Ethical Review Board in Linköping (Dnr: 2015/108-31). All participants received written information about the study and gave their written consent.

#### 2.1.3. Variables

Background variables collected pre-IMMRP and symptom-related self-reported variables collected at all three times (pre, post, and 12-month follow-up) were used in the analyses. The variables and instruments used are mandatory for the clinical specialist departments registering their data with the SQRP.

##### Independent Variables

This study focuses on the following variables:Sex (man or woman);Education level (university, upper secondary school, elementary school). In some of the analyses this variable was dichotomized and denoted as University (i.e., University versus the other alternatives);Country of birth—Sweden; other Nordic countries (i.e., Denmark, Finland, Iceland, and Norway); Europe except the Nordic countries; outside Europe. In some of the analyses, this variable was dichotomized as from Europe and outside Europe, labelled as “outside Europe”.

##### Other Variables Representing Clinical Presentations

For descriptive purposes, age (years) and self-reported days with no work or studies are given. For reports of the psychometric aspects of the measures below representing 22 specific outcomes, the reader is referred to other studies [26,49,50,51].

##### Pain Aspects

Average pain intensity during the previous seven days was registered using a 0–10 (0 = no pain and 10 = worst possible pain) numeric rating scale (NRS) (NRS-7days). In addition, pain duration (days) was registered.

##### The Multidimensional Pain Inventory (MPI)

MPI measures psychosocial, cognitive, and behavioral effects of chronic pain [52,53] and consists of three parts. The first part has five scales: Pain severity – measuring several aspects of the pain experience (MPI-Pain-severity); pain-related interference in everyday life (MPI-Pain-interfere); perceived life control (MPI-LifeCon); affective distress (MPI-Distress); and social support, i.e., perceived support from a spouse or significant others (MPI-SocSupp). The second part assesses the perception of responses to displays of pain and suffering from significant others: Punishing responses (MPI-Punish); solicitous responses (MPI-Solict); and distracting responses (MPI-Distract). The third part captures to what extent the patient participates in various activities using four scales. In the Swedish version of MPI, these scales are combined into a composite scale: the general activity index (MPI-GAI) [54].

##### Psychological Distress Variables

The Hospital Anxiety and Depression Scale (HADS) comprises seven items in each of two subscales: depression (HADS-D) and anxiety (HADS-A) symptoms [55,56]. Both subscale scores have a range of 0 to 21. A score of 7 or less in each subscale indicates a non-case, a score of 8–10 indicates a possible case, and a score of 11 or more indicates an almost definite case [55].

##### The Short Form Health Survey (sf36)

The Short Form Health Survey (sf36) addresses multidimensional health concepts and measurements of the full range of health states, including levels of well-being and personal evaluations of health [57]. The sf36 consists of eight dimensions with a scale from 0 to 100 where higher scores indicate a better perception of health [57]: (1) physical functioning (sf36-pf); (2) role limitations due to physical functioning (sf36-rp); (3) bodily pain (sf36-bp); (4) general health (sf36-gh); (5) vitality (sf36-vt); (6) social functioning (sf36-sf); (7) role limitations due to emotional problems (sf36-re); and (8) mental health (sf36-mh).

##### The European Quality of Life instrument (EQ-5D)

The European Quality of Life (EQ-5D) instrument captures a patient’s perceived state of health [58,59,60]. Five dimensions using a scale with three alternatives for each dimension were captured to obtain an index: mobility, self-care, usual activities, pain/discomfort, and anxiety/depression. This EQ-5D index is constructed based on a standardized valuation exercise where a representative sample of the general population in a country/region is asked to place a value on EQ-5D health states. The EQ-5D also measures self-estimation of today’s health according to a 100-point scale, a thermometer-like scale (EQ-VAS) with defined end points (high values indicate good health and low values indicate poor health).

#### 2.1.4. General Overall Outcomes of IMMRPs in This Study

##### Multivariate Improvement Score (MIS)

Most RCTs of IMMRPs use a substantial number of outcomes that requires special statistical considerations to deal with multiple comparisons and intercorrelations between outcomes. In a recent large study from SQRP investigating the 22 mandatory outcomes, 18 of the 22 outcomes demonstrated important intercorrelations [22]. The t-score of the first component (labelled MIS) of the advanced principal component analyses post-IMMRP and at 12-month follow-up represents a comprehensive measure of changes in mainly these 18 outcomes (see Appendix A). Hence, MIS is a relative overall outcome measure of IMMRP and higher MIS indicates an overall improvement; for details, see Ringqvist et al. [22]. The following MIS values were obtained for patients participating in IMMRPs:MIS post IMMRP: mean: −0.011 ± 2.59, 95% CI: −0.042–0.072, n = 14,666.MIS 12-m FU: mean: −0.011 ± 2.80, 95% CI: −0.054−0.068, n = 8851.(1)

To further illustrate the clinical importance of MIS, we performed a subgroup analysis (hierarchical clustering analysis; HCA) of MIS and identified three subgroups associated with large pairwise effect sizes between the three clusters [22]. At the 12-month follow-up, subgroup 1 had the highest MIS (5.01 ± 1.78, 95%CI: 4.90–5.11), subgroup 2 had the second highest MIS (0.78 ± 1.35; 95% CI: 0.74–0.82), and subgroup 3 had the lowest MIS (−2.43 ± 1.39, 95% CI: −2.47–−2.38) [22]. When scrutinizing the 22 outcomes, it was obvious that subgroup 1 generally showed clear improvements, subgroup 2 showed overall slightly positive improvements, and subgroup 3 showed no changes or deteriorations.

##### Estimations of Changes in Pain and in Life Situation

Post IMMRP and at the 12-month follow-up, the patients estimated the degree of positive change in pain (Change-pain) and in their ability to handle life situations in general (Change-life situation). Both items were rated on five-point Likert scales: Change-pain—markedly increased pain (0) to markedly decreased pain (4) and Change-life situation—markedly worsened (0) to markedly improved (4). The two variables were trichotomized (Change-pain—increased pain, no change, diminished pain and Change-life situation—worsened, no change, improved). Each scale had two positive and the two negative answering alternatives, respectively taken together.

### 2.2. Statistics

All statistics were performed using the statistical package IBM SPSS Statistics (version 24.0; IBM Corporation, Route 100 Somers, New York, NY, USA). A probability of <0.001 (two-tailed) was accepted as the criteria for significance due to the large number of subjects. Text and tables report the mean value ± one standard deviation (± 1 SD) of continuous variables. Percentages (%) are reported for categorical variables. SQRP uses predetermined rules when handling single missing items of a scale or a subscale; details are reported elsewhere [61]. To compare groups, we used student’s t-test for independent samples, analysis of variance (ANOVA; Bonferroni post hoc test if significant difference), and Chi square test. Effect sizes (ES; Cohen’s d) for within group analysis were computed using a calculator when appropriate. Hedges’g, the measure of effect size weighted according to the relative size of each sample, was used for between group ES using a calculator. The absolute effect size was considered very large for ≥1.3, large for 0.80–1.29, moderate for 0.50–0.79, small for 0.20–0.49, and insignificant for <0.20 [62]. A recent article of ours presents how MIS was obtained [22]. (Statistical details are given in Appendix A).

To understand how the sociodemographic variables taken together (i.e., sex, education, and country of birth) influenced the clinical presentation at baseline and outcomes of IMMRP, we performed a two-step cluster analysis using sex, education, and country of birth as input variables to identify clusters (i.e., log-likelihood measure distance, number of clusters determined automatically, and Schwartz’s Bayesian cluster criterion were options). To obtain reasonably large clusters, the ratio between clusters sizes had to be <3.0, per the convention for this analysis. We predetermined that 3–5 clusters were optimal and chose the number of clusters that optimized cluster quality (i.e., silhouette measure of cohesion and separation) and was above 0.5 (good). A predictor importance >0.60 for included variables was another requirement.

## 3. Results

As reported elsewhere, there were 39,916 chronic pain patients registered in the SQRP database and most patients (76.3%) were women [26]. Of these, 14,666 patients participated and completed the SQRP questionnaire before and on at least one of the two time points after the IMMRP [22]; 60% of the patients answering the questionnaires pre-IMMRP and post-IMMRP also answered the questionnaires at the 12-month follow-up.

### 3.1. Sex

#### 3.1.1. Baseline Situation—Total Database

A small significant difference in age was found (insignificant ES) (Table 1). Women (21.9% elementary school, 52.3% upper secondary school, and 25.8% university) had somewhat higher education levels than men (24.6% elementary school, 56.9% upper secondary school, and 18.4% university) (Chi^2^ = 237.8, df = 2, *p* < 0.001). A significantly larger proportion of men (73.8% Sweden, 2.5% other Nordic country, 6.1% Europe outside Nordic countries, and 17.6% outside Europe) than women (79.1% Sweden, 2.8% other Nordic country, 5.4% Europe outside Nordic countries, and 12.7% outside Europe) were born outside Europe (Chi^2^ = 170.2, df = 3, *p* < 0.001). No sex difference existed for days with no work or studies. Pain duration was significantly longer in women than men (insignificant ES) (Table 1).

At baseline, several significant sex differences existed in the total SQRP database for the variables used as repeated measures. For example, women reported higher pain intensity, higher severity, lower social support, and worse situations on most of the subscales of the sf36 (Table 1).However, men reported more depressive symptoms, perceived more punishing responses, worse quality of life according to EQ-5D, and were less active (Table 1). Hence, no consistent sex pattern emerged. The ES for these sex differences were clinically insignificant with one exception: women reported a worse situation according to the vitality scale of the sf36 (a small ES, i.e., ES = 0.22).

#### 3.1.2. Sex Differences in Overall and Specific Outcomes of IMMRP

A significantly higher proportion of women than men participated in IMMRP (Women: 38.4% vs. Men: 30.8%; Chi^2^ = 207.9, df = 1; *p* < 0.001). No significant sex difference was observed for MIS post-IMMRP (Table 2). At the 12-month follow-up, the MIS showed overall significantly better results in women (Women: 0.03 ± 2.77 vs. Men: −0.16 ± 2.90; *p* = 0.008, ES = 0.07), but ES indicated that this was a clinically insignificant difference (Table 2).

For the specific 22 outcome variables, a few significant sex differences were noted with respect to changes from baseline to post-IMMRP (Appendix A). Women reported larger changes in vitality, general health, and depressive symptoms. However, these differences were clinically insignificant (ES < 0.20). At the 12-month follow-up, women reported significantly larger changes in depressive and anxiety symptoms and in general health than men (Appendix A). However, these sex differences were clinically insignificant (ES < 0.20).

No significant sex difference existed post-IMMRP for change-pain, but a larger proportion of women than men reported improvements for change-life situation (84.7% vs. 80.5%) (Table 2). For these two items, a similar situation was found at the 12-month follow-up (Table 2).

### 3.2. Education

#### 3.2.1. Baseline Situation: Total Database

At baseline, significant differences existed for all 22 variables: patients with only elementary school reported the worst situation, patients with upper secondary school reported an intermediary situation, and patients with university education reported the best situation (Table 3). The majority of pairwise ES between elementary school and university were small (ES: 0.20–0.49). The highest ES were noted for pain intensity aspects (NRS-7d) (ES = 0.43), MPI-Pain severity (ES = 0.47), and physical functioning (sf36-pf; ES = 0.38).

#### 3.2.2. Education: Differences in Overall and Specific Outcomes of IMMRP

Elementary school education level was associated with significantly lower participation in IMMRP (elementary school: 31.6%, upper secondary school: 38.9%, university: 39.4%; Chi^2^ = 164.2, df = 2; *p* < 0.001). Both MIS variables revealed significant differences between education levels. Hence, outcomes were best in those with university education and worst in those with elementary school education (Table 4). However, the pairwise comparisons (elementary school vs. university) revealed that the ES were insignificant (<0.20) and small (ES: 0.20–0.49) at post-IMMRP (ES = 0.13), and at the 12-month follow-up (ES = 0.22). A similar pattern for MIS was obtained for the two change variables. At both timepoints, those with elementary school reported improvements in change-pain to a lesser extent: post-IMMRP (elementary school: 52.9%; upper secondary school: 55.6%; University: 61.8%) (Chi^2^ = 57.0, df = 4, *p* < 0.001) and at the 12-month follow-up (elementary school: 48.1%; upper secondary school: 56.4%; University: 63.0%) (Chi^2^ = 93.7, df = 4, *p* < 0.001).

The changes in 8 of the 22 specific outcome variables showed significant differences post-IMMRP (Appendix A) but these were clinically insignificant (ES < 0.20). At the 12-month follow-up the changes in 13 variables showed significant differences with respect to education level (Appendix A) and four of these were associated with small ES (ES < 0.20), i.e., patients with university education had larger changes than patients with elementary school education in pain severity, pain interference, physical function, and social function.

For the change-life situation, variable differences related to education level were also detected. Those with elementary school reported fewer improvements in their situation both post-IMMRP (elementary school: 78.7%; upper secondary school: 83.8%; university: 88.1%) (Chi^2^ = 95.4, df = 4, *p* < 0.001) and at the 12-month follow-up (elementary school: 68.3%; upper secondary school: 77.3%; University: 81.2%) (Chi^2^ = 94.8, df = 4, *p* < 0.001).

### 3.3. Country of Birth

#### 3.3.1. Baseline Situation—Total Database

The reported situation was overall worse for those born outside Europe. When comparing patients born in Europe with those born outside Europe, all measures at baseline were significant and the ESs were generally at least small (Table 5). Pain intensity aspects, psychological distress variables, MPI-protect, general health, and role emotional of sf36 were associated with medium ES.

#### 3.3.2. County of Birth—Differences in Overall and Specific Outcomes of IMMRP 

Patients from outside Europe had the lowest participation in IMMRP (Sweden: 39.1%; other Nordic countries: 37.9%; Europe except the Nordic countries: 34.3%; outside Europe: 28.0%; Chi^2^ = 257.8, df = 3; *p* < 0.001).

For MIS, no significant differences existed post-IMMRP. Significant differences in MIS were noted at the 12-month follow-up with respect to country of birth. The post hoc analysis showed that patients born in Sweden had significantly better outcomes than those born in Europe (outside Nordic countries) and outside Europe (Table 6). However, pairwise ESs were clinically insignificant, i.e., Sweden vs. Europe (ES = 0.16) and Sweden vs. outside Europe (ES = 0.11).

Only few significant differences in changes of the 22 outcome variables between patients from Europe vs. outside Europe existed and they were all clinically insignificant (ES < 0.20) (Appendix A).

At the 12-month follow-up, but not the post-IMMRP, significant differences were recorded for change-pain (i.e., improvements in change-pain): Sweden: 57.8%, other Nordic country: 53.2%, Europe outside Nordic countries: 46.5%, and outside Europe: 50.6% (Chi^2^ = 68.5, df = 6, *p* < 0.001). For change-life situation, significant differences were also found for the country of birth both at the post-IMMRP and the 12-month follow-up. Improvements in change-life situation had the following distribution post-IMMRP: Sweden: 84.9%; other Nordic country: 86.1%; Europe outside Nordic countries: 80.7%; and outside Europe: 77.3% (Chi^2^ = 61.7, df = 6, *p* < 0.001). At the 12-month follow-up, the corresponding figures were as follows: Sweden: 78.4%; other Nordic country: 76.5%; Europe outside Nordic countries: 66.8%; and outside Europe: 65.5% (Chi^2^ = 110.2, df = 6, *p* < 0.001). Hence, outcomes were better for those born in Sweden than those born outside Europe. 

### 3.4. Clusters Based On the Sociodemographic Variables

#### 3.4.1. Baseline Situation

The most optimal solution for the two-step cluster analyses was five clusters with sex, education level, and country of birth (dichotomized) as input variables (Table 7). As intended, input variables differed significantly between the five clusters. Three clusters with women, born in Europe and with different education levels were identified (clusters 1–3). Cluster 4 was formed by men born in Europe with different education levels. Patients of the 5th cluster were born outside Europe, included both sexes and with mixed education levels.

Significant differences between the clusters existed for all variables at baseline (all *p* < 0.001) (Table 8); cluster 1 (women born in Europe with university education) and cluster 5 (born outside Europe and with different education levels) showed the most marked differences (ES: moderate or large). Cluster 2 (women in Europe with elementary school) generally showed the second worst situation after cluster 5 and on all except two variables differed significantly from cluster 1 according to the post hoc tests. In the clusters with only women (clusters 1–3), an inverse relationship between education level and clinical severity variables in Table 8 were found.

#### 3.4.2. Clusters: Differences in Overall Outcomes of IMMRP 

At baseline, 37.4% of the patients participated in IMMRPs. Women with secondary upper school (cluster 3) and university education (cluster 1) had higher proportions participating in IMMRPs than the three other clusters (Table 7). The lowest proportion of participation was found in cluster 5 (i.e., born outside Europe and with mixed education levels).

Significant differences in MIS across the five clusters were noted post-IMMRP and at 12-month follow-up (Table 9). Cluster 1 (European women with university education) benefited most from IMMRP, while cluster 2 (European women with elementary school) and cluster 5 (born outside Europe and with different education levels) benefited the least. At the 12-month follow-up, ESs were small according to the comparisons of cluster 1 versus cluster 2 and cluster 5.

Diminished pain (i.e., change-pain) was highest in European women with university education (cluster 1) and lowest in European women with elementary school (cluster 2) and in those born outside Europe and with different education levels (cluster 5) (Table 9). For the change-life situation variable, cluster 1 reported the best situation and cluster 5 the worst situation at both time points (post-IMMRP: 88.6% vs. 77.3% and at 12-month follow-up: 83.5% vs. 65.7%) (Table 9).

## 4. Discussion

The most interesting results of this large PROM study from SQRP were generated from the cluster analysis identifying five clusters as prominent significant differences in the baseline clinical situation. The best baseline clinical situation was found in cluster 1 (European women with university education) and the worst situation in cluster 5 (born outside Europe of both sexes and different education levels) (pairwise ESs: moderate or large). Moreover, European women with university education (cluster 1) also had the most favorable overall outcomes in response to IMMRPs—i.e., MIS (small ES at 12-month FU between some clusters)—and in global estimations of perceived change in pain and ability to handle life situations (change-pain and change-life). Moreover, within the clusters with only women (cluster 1-3), an inverse relationship was found between education level and clinical severity as well as overall outcomes of IMMRP.

When analyzed separately, each factor (sex, education level, and country of birth) generally revealed insignificant or small magnitudes of differences in clinical presentations according to ES. Such analysis of overall outcomes of IMMRPs also showed significant differences with respect to sex, education level, and country of birth, but these differences were generally associated with insignificant ESs.

### 4.1. Sex Aspects

The prevalence of chronic pain is higher in women than men [63,64,65]. It is unclear whether sex differences for pain severity exist [66,67,68] as some studies report greater pain severity in women and other studies report no sex differences. It has been proposed that methodological factors may influence results [66]. In a SQRP study from one university clinical department, no sex differences in pain severity were found [50]. However, women reported significantly higher activity level, satisfaction with life situation, satisfaction with sexual life, pain acceptance, and social support, whereas men reported higher degree of kinesiophobia, mood disturbances, and lower activity level [50]. ESs were insignificant or small. In the present considerably larger study, women reported significantly higher pain intensity than men, but with insignificant ES in the pairwise comparison. No consistent sex pattern in the investigated variables emerged at baseline and ESs were with one exception, insignificant (i.e., <0.20) (Table 1). For vitality, which is associated with a small ES, a lower value was found for women than men. Small significant sex differences in background variables such as country of birth and in pain-related factors such as pain duration variables were also found (Table 1). Hence, we did not find prominent sex differences.

Women are overrepresented in specialist clinics in Sweden compared to the community prevalence and this selection is increased when analyzing the proportion who participated in IMMRP [26,27]. Hence, in the present study, 76.3% of the patients registered in SQRP were women. We could not confirm earlier reports that women assessed at specialist clinics have a more severe clinical situation than men [50] or are judged to be more prone to behavioral change. We concluded that clinical presentations cannot readily explain why men take part in IMMRP to somewhat lesser extent than women. The reasons for this skewed assortment/selection are unclear and need to be addressed. Adherence to the biopsychosocial model does not exclude unconscious motives such as how the perceived quality of the doctor-patient interaction can influence whether perceived symptoms are more or less explained by biological factors and could as such have an impact on treatment recommendations [69].

Few large studies have examined the sex differences in outcomes of IMMRPs. The existing literature is conflicting: women benefit more [68,70,71], no sex differences [43,72,73], and men benefit more [74,75]. The outcomes of IMMRPs in a primary care SQRP study were better in women than in men [76]. The overall outcome variable MIS in this study showed the same pattern at 12-month follow-up, i.e., significantly better results for women but ES did not display clinical importance. Women were more positive in their retrospective assessment of how much their pain coping skills (i.e., change-pain and change-life situation) had improved in response to their participation in the IMMRPs. Hence, in our large-scale real-life scenario we do not find evidence for substantial sex differences in outcomes. The conflicting results in the literature may be due to different cohorts investigated as well as the choice of outcomes. Because IMMRP is a complex intervention, we used three overall outcomes. A question is if the range of activities in IMMRP and the manner they are presented appeal equally to both sexes?

### 4.2. Education

Significant differences in clinical presentations existed at baseline (Table 3). The most prominent differences were found between those with elementary school education and those with university education; most variables displayed a small ES for the pairwise comparisons. Level of education can also be a proxy for socioeconomic position. Our results agree with studies reporting that prevalence of chronic pain, severity of pain, and disability are inversely related to socio-economic position [29,30,77,78]. For example, a review found that social contexts were seldom considered in studies of IMMRP and concluded that social circumstances should be given increased consideration [79].

MIS as well as the two retrospective variables showed significant differences between the levels of education at both timepoints (Table 4); elementary school education was associated with less improvements than university education. However, at the 12-month follow-up, ES for MIS was small for the pairwise comparison elementary school vs. university. The reasons for this difference in overall outcomes (MIS and retrospective variables) between elementary school education and university education—as well as the lower participation rate for patients with elementary school education level—are unclear and requires further research. It could reflect differences in overall severity but could also be related to the content of IMMRP.

### 4.3. Country of Birth

There are substantial and complex ethnic variations in prevalence and outcomes of pain conditions [77,80]. Patients born outside Europe reported a significantly worse situation in clinical presentations at baseline than those born in Europe; generally, ESs were small (Table 4). This finding is in agreement with other studies (e.g., non-western born immigrants residing in Sweden experienced a greater impact of chronic pain than their Swedish-born counterparts) [38]. In another study, immigrants reported a higher pain prevalence and higher pain intensity than natural born inhabitants [39]. The reasons for this picture at baseline are probably complex. Some of the patients born outside Europe have fled from war and war-like situations and may live with physical and psychological burdens. Moreover, an interaction effect between country of birth and education level may be present since a larger proportion of patients from outside Europe had elementary school education compared to those born in Europe (28.4% vs. 21.7; Chi^2^ = 158.5, df = 2). Hence, socioeconomic factors (see above) may also influence. On the other hand, patients born outside Europe may represent a selection of non-Europeans in the sense that they have answered the Swedish questionnaire of SQRP and therefore are relatively fluent in Swedish and well-integrated in the Swedish society; if such a selection influences the results are unclear.

There are reports from small cohorts that immigrants benefit less from IMMRPs than native patients [38,40,41,42,43]. These results were challenged in a small study reporting no differences in IMMRPs outcomes [44]. In this large study, we found significant differences in MIS after IMMRPs at the 12-month follow-up (Table 6), but the ESs for MIS at the two timepoints were insignificant. A similar pattern was noted for change-pain and change-life situation with fewer improvements in patients born outside Europe. A review suggested that in particularly non-Western backgrounds may be associated with other attitudes towards self-management interventions, passive symptom-focused management strategies, as well as pharmacological treatments [80], which may influence IMMRP outcomes. IMMRPs may not meet the needs of patients outside Europe. Another alternative is that participation in (lower in non-European patients) and IMMRP outcomes are hampered by different biases of the professionals/team towards non-European patients and/or insufficient knowledge about immigration and other cultures.

### 4.4. Clusters Based On Sex, Education Level, and Country of Birth

Clinical presentations showed marked differences (at least small ESs) with respect to education level and country of birth. As standalone factors, these are important to consider in the assessments of chronic pain patients. More importantly, certain combinations of education level and country of birth factors together with sex need to be considered in the assessment procedures according to the cluster analysis. Hence, the cluster analysis was made to investigate if certain combinations of sex, education level, and country of birth were associated with positive or negative IMMRP outcomes. The importance of this is clearly illustrated for the investigated variables at baseline (Table 8). Generally marked differences in PROMs (ESs: moderate or large) existed between European women with university education (cluster 1) and those born outside Europe with different education levels (cluster 5). Relatively marked differences were also noted for the female European clusters (clusters 1–3), indicating the importance of education level.

Although patients born outside Europe (cluster 5) report a more severe clinical picture (Table 6), they participate to a less extent in IMMRP than European women with university education (cluster 1) (Table 5). European women with elementary school education (cluster 2) participated less than European women with university education (cluster 1) (Table 7), results that agree with another study [30]. We have earlier reported a higher female participation [26] and this is also confirmed in the cluster analysis (Table 7). The reasons for these differences in participation rate still need to be investigated. Patients may have chosen not to participate for various reasons; mapping of the perceived barriers may give important clues as to how participation for patients belonging to clusters with the most prominent clinical severity can be improved. Treatment recommendations of experienced interdisciplinary teams may be influenced by discriminatory attitudes and sex bias [81]; e.g., preconceptions about which patients fit into IMMRPs, which patients may benefit, and the importance of sociodemographic factors (e.g., age, sex, education level, place of origin, ethnicity, verbal skills, and social class) for positive outcomes of IMMRP [82]. Increasing the providers’ awareness of social and cultural aspects when assessing and delivering IMMRP have been discussed as necessary [43].

The clusters based on the three factors showed significant differences in overall outcomes and with small ES at the 12-month follow-up for MIS. The best results according to the overall outcomes at the 12-month follow-up were seen in European women with university education (cluster 1) and the worst in European women with elementary school education (cluster 2) and in the non-European cluster (cluster 5). The specific reasons why female patients with elementary school education and patients born abroad had lower participation in IMMRP and poorer results may speculatively be due to that a lower level of education indicates more stressful working conditions and lower salary (which in turn makes sick leave more difficult). Other factors may be that a lower level of education or difficulties with the Swedish language means greater difficulties in analyzing and implementing self-help advice and changing non-appropriate lifestyle habits. We have previously reported that for this cohort, those with the worst clinical picture report the largest changes after IMMRPs [22,26,83]. This is true on a general level, but the picture evidently becomes more complicated when incorporating sociodemographic factors such as sex, education, and country of birth. In future analyses it will be important to understand the relative importance of the severity of the clinical presentation and the different combinations of the variables sex, education level, and country of birth for the overall outcomes of IMMRP. This may give important clues how to improve the outcomes of IMMRP. The fact that both clusters 4 and 5 included patients with different educational backgrounds should lead to in-depth analyzes of the outcomes within these clusters in future studies.

As discussed elsewhere, effect sizes for IMMRP surpass pharmacological treatments [84,85,86,87,88,89], but they are still small to moderate [11,12,18,22,90] and efforts should be invested into constant improvement. An important principle in healthcare is equity (i.e., prioritization of healthcare based on the need of the patient), but low education, male sex, and/or non-European origin appear to be associated with lower participation rates and worse results of IMMRP, suggesting that equal care is not delivered. Carr and Moffet asked provocatively whether CBT interventions designed by middle class health professionals are more suitable for middle class patients [29]. Our results indicate that the outcomes may be improved if the combination of sex, education, and country of birth are considered when assessing chronic pain patients and when designing IMMRPs.

### 4.5. Clinical Implications

The combination of sex, education level, and country of birth needs to be considered in the clinical assessment of patients with chronic pain. Hence, patients from outside Europe (independent of sex and education level; cluster 5) generally had a worse situation than women from Europe with university education (cluster 1). Despite a more clinically severe situation the patients from outside Europe (cluster 5) participated less in IMMRP than women with higher education levels. Moreover, the male group (cluster 4) and women with low education (cluster 2) had lower participation. In the perspectives of fairness and equality the probably complex reasons for these differences must be further analyzed both in clinical practice and in research. Moreover, there is a need to understand the reasons for the worse overall outcomes of IMMRP in patients from outside Europe (cluster 5), in males (cluster 4) and in women with low education level (cluster 2). The importance of the clinical presentation, the content and delivery of IMMRP and other factors are important to consider improving outcomes.

### 4.6. Strengths and Limitations

The large cohort of chronic pain patients with a nation-wide representation is a strength, but the cohort represents a selection of the most complex cases and cannot simply be generalized to primary care settings. The large number of patients used in the analyses pinpoints the need to determine whether significant differences are clinically important when using ES. Most studies referred to above are considerably smaller, have focused on statistical differences, and are not based on the non-selected flow of chronic patients in practice settings. However, the large number of patients in the present study does not exclude random statistical differences between groups which in turn may be associated with clinical differences according to ES. Although validated and well-known PROM instruments were used, these may be problematic in repeated evaluations [91]. Changes that patients undergo because of IMMRPs may affect the interpretations of the PROM questions when presented post-IMMRPs and at follow-up. Retrospective evaluations may be problematic (e.g., recall time, desirability, memory aspects) [92,93,94]. On the other hand, MIS and these two items generally showed the same pattern. Although all specialist clinics’ IMMRPs can be included in the general description of IMMRP (see introduction), there may be heterogeneity regarding scope and intensity of the different IMMRP components, as well as different competence of the therapists in the team. Furthermore, the team’s internal interaction, interaction with the patient and interaction with other relevant actors (e.g., the employers and the Social Insurance Agency) can differ. At present, no detailed information is available within the registry that captures these aspects, which is a limitation. No control condition was available, which ethically is complicated to arrange for a registry of real-life practice patients.

## 5. Conclusions

This large-scale study of IMMRPs real life clinical settings demonstrates significant differences in clinical presentations and overall outcomes at the 12-month follow-up with respect to sex, education, and country of birth. However, the clinical importance with respect to different levels/categories of these variables according to effect sizes were generally insignificant or small. Clusters based on sex, education, and country of birth showed marked differences in clinical presentations (moderate or large effect sizes) between some of the clusters as well as differences in outcomes. These results raise important questions concerning fairness and equality. The combination of sex, education, and country of birth needs to be considered in the assessment of chronic pain patients. These factors are also important to consider when optimizing the content and delivery of IMMRP in clinical practice.

## Figures and Tables

**Table 1 jcm-09-02374-t001:** Background and baseline data for variables used in the repeated analyses for all subjects by sex (Mean ± SD, together with n) in the total Swedish Quality Registry for Pain Rehabilitation (SQRP) database. Note that a subgroup of these patients participated in the IMMRP. Group comparison (*t*-test) and effect size (ES; Hedges’g) are to the far right.

Sex	Women			Men			Statistics	
Variables	n	Mean	SD	n	Mean	SD	*t*-test	ES
Age (years)	29,318	42.73	11.35	11,383	44.30	11.36	<0.001	0.14
Days with no work/studies (days)	11,056	1383	2402	4407	1274	2351	0.010	0.04
Pain duration (days)	25,165	3150	3292	9928	2977	3359	<0.001	0.05
NRS-7d	27,906	7.10	1.74	10,705	6.88	1.86	<0.001	0.12
HAD-A	28,277	9.25	5.02	10,845	9.20	5.01	0.326	0.01
HAD-D	28,290	8.59	4.66	10,850	9.02	4.89	<0.001	0.09
MPI-Pain sever	28,081	4.52	0.93	10,766	4.38	1.01	<0.001	0.15
MPI-Pain interfere	27,807	4.42	1.07	10,652	4.40	1.11	0.131	0.02
MPI-control	27,964	2.64	1.16	10,718	2.66	1.21	0.332	0.02
MPI-distress	28,003	3.51	1.33	10,714	3.50	1.36	0.441	0.00
MPI-SOCSupp	27,907	4.12	1.41	10,675	4.29	1.37	<0.001	0.12
MPI-punish	25,765	1.72	1.42	9484	1.93	1.37	<0.001	0.15
MPI-protect	25,644	3.03	1.47	9426	2.99	1.42	0.010	0.03
MPI-distract	25,726	2.55	1.25	9462	2.59	1.23	0.007	0.03
MPI-GAI	27,980	2.39	0.88	10,683	2.25	0.97	<0.001	0.15
EQ-5D-index	27,082	0.24	0.31	10,463	0.21	0.32	<0.001	0.10
EQ-VAS	26,553	40.35	19.97	10,284	40.45	21.07	0.681	0.00
sf36-pf	27,578	50.12	21.90	10,593	52.31	23.60	<0.001	0.10
sf36-rp	27,061	13.16	25.40	10,330	14.13	26.92	0.001	0.04
sf36-bp	27,597	23.10	14.95	10,599	24.22	15.80	<0.001	0.07
sf36-gh	27,227	39.22	21.08	10,440	41.40	20.67	<0.001	0.10
sf36-vt	27,515	22.94	18.72	10,559	27.11	20.18	<0.001	0.22 *
sf36-sf	27,595	45.83	25.95	10,582	47.41	27.19	<0.001	0.06
sf36-re	26,702	42.70	43.39	10,156	39.97	43.11	<0.001	0.06
sf36-mh	27,487	54.25	22.35	10,542	52.72	23.54	<0.001	0.07

* = small effect ES; NRS-7d = pain intensity previous 7 days; HAD = hospital anxiety and depression scale; HAD-A = subscale anxiety; HAD-D = subscale depression; MPI = multidimensional pain inventory; MPI-pain-sever = subscale pain severity; MPI-pain-interfere = subscale pain related Interference; MPI-control = subscale life control; MPI-distress = subscale affective distress; MPI-SOCSupp = subscale social support; MPI-punish = subscale punishing responses; MPI-protect = subscale solicitous responses; MPI-distract = subscale distracting responses; MPI-GAI = subscale general activity index; EQ = European quality of life instrument; EQ-5D-index = index based om five dimensions; EQ-VAS = self-estimation of health; sf36 = short form health survey; sf36-pf = physical functioning; sf36-rp = role limitations due to physical functioning; sf36-bp = bodily pain; sf36-gh = general health; sf36-vt = vitality; sf36-sf = social functioning; sf36-re = role limitations due to emotional problems; sf36-mh = mental health.

**Table 2 jcm-09-02374-t002:** Outcomes of interdisciplinary multimodal pain rehabilitation programs (IMMRPs) according to a multivariate improvement score (MIS) and change variables (change-pain and change-life) in women and men post-IMMRP and at 12-month follow up. Statistics are furthest to the right, i.e., group comparison (*t*-test) together with effect size (ES; Hedges’g) and Chi^2^ test.

Sex	Women			Men			Statistics	ES
Variables	*N*	Mean	SD	*N*	Mean	SD	*p*-value	
MIS								
MIS post-IMMRP	11,183	0.01	2.59	3483	−0.08	2.56	0.075	0.03
MIS 12-month FU	6822	0.03	2.77	2029	−0.16	2.90	0.008	0.07
		%			%		*p*-value (Chi^2^ test)	Chi^2^; df
Pain-change (% diminished pain)								
Post IMMRP	10,487	56.4		3270	57.2		0.279	2.55; 2
12-month FU	6590	56.3		1966	57.1		0.048	6.06; 2
Life-change (% improved)								
Post-IMMRP	10,543	84.7		3275	80.5		<0.001	32.5; 2
12-month FU	6602	78.3		1968	70.9		<0.001	48.0; 2

Chi^2^ = Chi square; df = degrees of freedom; FU = follow-up; ES = effect size.

**Table 3 jcm-09-02374-t003:** Comparison at baseline between different levels of education. The statistics (ANOVA including post hoc tests) and effect size (ES; Hedges’ g) for Elementary School vs. University are to the far right.

Education Level	Elementary	School		Upper	Secondary	School	University			ANOVA		ES
Variables	*N*	Mean	SD	*N*	Mean	SD	*N*	Mean	SD	*p*-value	post-hoc	
Age (years)	8807	44.55	12.22	20,840	41.85	11.20	9256	44.61	10.49	<0.001	Esc = U; other different	0.01
Days no work/studies	3991	1671	2518	8464	1278	2389	2837	1121	2171	<0.001	all different	0.23 *
Pain duration (days)	7559	3298	3408	18,711	3003	3173	8365	3126	3498	<0.001	all different	0.05
NRS-7d	8529	7.40	1.72	20,439	7.07	1.72	9059	6.62	1.88	<0.001	all different	0.43 *
HAD-A	8617	9.85	5.09	20,517	9.24	5.00	9147	8.66	4.91	<0.001	all different	0.24 *
HAD-D	8622	9.20	4.73	20,527	8.73	4.74	9155	8.22	4.64	<0.001	all different	0.21 *
MPI-Pain sever	8482	4.68	0.92	20,471	4.51	0.92	9071	4.23	1.01	<0.001	all different	0.47 *
MPI-Pain interfere	8367	4.54	1.04	20,292	4.45	1.05	9005	4.24	1.15	<0.001	all different	0.27 *
MPI-control	8436	2.51	1.21	20,385	2.65	1.17	9052	2.77	1.14	<0.001	all different	0.22 *
MPI-distress	8453	3.63	1.36	20,404	3.53	1.33	9049	3.34	1.32	<0.001	all different	0.22 *
MPI-SOCSupp	8426	4.28	1.41	20,350	4.21	1.38	8996	3.97	1.39	<0.001	all different	0.22 *
MPI-punish	7628	1.83	1.43	18,636	1.78	1.40	8282	1.71	1.39	<0.001	Esc = USS, other different	0.09
MPI-protect	7588	3.17	1.51	18,541	3.05	1.46	8242	2.80	1.38	<0.001	all different	0.26 *
MPI-distract	7616	2.63	1.29	18,605	2.59	1.23	8267	2.44	1.21	<0.001	Esc = USS, other different	0.15
MPI-GAI	8447	2.23	0.93	20,380	2.37	0.90	9035	2.40	0.89	<0.001	USS = U, other different	0.19
EQ-5D-index	8157	0.19	0.30	19,680	0.23	0.31	8847	0.28	0.32	<0.001	all different	0.29 *
EQ-VAS	7957	38.12	20.59	19,341	40.37	20.08	8726	42.38	20.11	<0.001	all different	0.21 *
sf36-pf	8401	45.98	22.09	20,041	51.11	22.02	8969	54.49	22.71	<0.001	all different	0.38 *
sf36-rp	8128	12.58	25.33	19,679	13.44	25.68	8857	14.13	26.47	<0.001	Esc NE U	0.06
sf36-bp	8401	21.27	15.06	20,076	23.12	14.68	8960	25.99	15.99	<0.001	all different	0.30 *
sf36-gh	8250	36.42	20.30	19,814	40.05	20.77	8863	42.40	21.63	<0.001	all different	0.28 *
sf36-vt	8370	22.96	18.88	20,006	23.85	19.04	8943	25.49	19.82	<0.001	all different	0.13
sf36-sf	8386	45.32	26.46	20,061	46.68	26.23	8973	46.06	26.27	<0.001	Esc = U, other different	0.03
sf36-re	7971	37.25	42.66	19,417	41.81	43.24	8771	46.88	43.61	<0.001	all different	0.22 *
sf36-mh	8352	51.30	23.04	19,984	53.82	22.76	8945	56.20	21.98	<0.001	all different	0.22 *

Esc = Elementary School; USS = upper secondary school; U = University; NE = not equal; * = small effect ES; NRS-7d = pain intensity previous 7 days; HAD = hospital anxiety and depression scale; HAD-A = subscale anxiety; HAD-D = subscale depression; MPI = multidimensional pain inventory; MPI-pain-sever = subscale pain severity; MPI-pain-interfere = subscale pain related Interference; MPI-control = subscale life control; MPI-distress = subscale affective distress; MPI-SOCSupp = subscale social support; MPI-punish = subscale punishing responses; MPI-protect = subscale solicitous responses; MPI-distract = subscale distracting responses; MPI-GAI = subscale general activity index; EQ = European quality of life instrument; EQ-5D-index = index based om five dimensions; EQ-VAS = self-estimation of health; sf36 = short form health survey; sf36-pf = physical functioning; sf36-rp = role limitations due to physical functioning; sf36-bp = bodily pain; sf36-gh = general health; sf36-vt = vitality; sf36-sf = social functioning; sf36-re = role limitations due to emotional problems; sf36-mh = mental health.

**Table 4 jcm-09-02374-t004:** MIS by education level post-IMMRP (n = 14,439) and at 12-month follow-up (n = 8706). ANOVA and post-hoc tests are to the far right.

	Elementary School			Upper Secondary School			University			ANOVA	
Variables	*N*	Mean	SD	*N*	Mean	SD	*N*	Mean	SD	*p*	post hoc
MIS at post IMMRP	2746	−0.19	2.55	8055	−0.02	2.57	3638	0.15	2.62	<0.001	Esc NE U (*p* < 0.001); Esc NE Uss (*p* = 0.010); Uss NE U (*p* = 0.002)
MIS at 12-month FU	1666	−0.36	2.70	4809	−0.01	2.78	2231	0.26	2.86	<0.001	all different

FU = follow-up; Esc = Elementary school; U = university; Uss = Upper secondary school; NE = not equal.

**Table 5 jcm-09-02374-t005:** Comparison at baseline between those born in Europe and those born outside Europe. The statistics (*t*-test) and effect size (ES; Hedges’g) are to the far right.

Country of Birth	Europe			Outside Europe			Statistics	ES
Variables	*N*	Mean	SD	*N*	Mean	SD	*p*-value	
Age (years)	33,784	42.92	11.65	5528	44.42	9.41	<0.001	0.13
Days no work/studies	13,091	1374	2431	2323	1230	2141	0.007	0.06
Pain duration (days)	30,062	3188	3378	4870	2565	2818	<0.001	0.19
NRS-7d	32,989	6.88	1.76	5386	7.95	1.62	<0.001	0.61 **
HAD-A	33,298	8.68	4.82	5336	12.72	4.79	<0.001	0.84 ***
HAD-D	33,317	8.38	4.64	5338	10.82	4.72	<0.001	0.52 **
MPI-Pain sever	33,115	4.41	0.95	5261	4.94	0.86	<0.001	0.56 **
MPI-Pain interfere	32,902	4.37	1.09	5107	4.75	0.94	<0.001	0.35 *
MPI-control	33,037	2.71	1.16	5185	2.26	1.23	<0.001	0.38 *
MPI-distress	33,060	3.39	1.32	5193	4.21	1.20	<0.001	0.63 **
MPI-SOCSupp	32,922	4.13	1.39	5196	4.41	1.39	<0.001	0.20 *
MPI-punish	30,283	1.74	1.38	4576	2.02	1.53	<0.001	0.20 *
MPI-protect	30,139	2.90	1.42	4544	3.80	1.50	<0.001	0.63 **
MPI-distract	30,236	2.48	1.20	4567	3.07	1.40	<0.001	0.48 *
MPI-GAI	33,022	2.41	0.86	5186	1.97	1.10	<0.001	0.49 *
EQ-5D-index	31,875	0.25	0.31	5136	0.12	0.30	<0.001	0.42 *
EQ-VAS	31,410	41.01	20.00	4926	36.20	21.49	<0.001	0.24 *
sf36-pf	32,621	52.04	22.16	5124	42.58	22.15	<0.001	0.43 *
sf36-rp	32,182	13.72	25.96	4798	11.42	24.70	<0.001	0.09
sf36-bp	32,642	24.27	15.07	5124	17.86	14.66	<0.001	0.43 *
sf36-gh	32,288	41.34	20.84	4964	29.83	19.08	<0.001	0.56 **
sf36-vt	32,557	24.19	19.25	5090	23.24	18.98	<0.001	0.05
sf36-sf	32,639	47.49	26.32	5110	38.36	24.79	<0.001	0.35 *
sf36-re	31,878	44.74	43.55	4586	22.98	36.48	<0.001	0.51 **
sf36-mh	32,534	55.91	22.06	5073	40.56	22.26	<0.001	0.69 **

* = small effect ES, ** = medium effect ES, *** large effect ES); NRS-7d = pain intensity previous 7 days.

**Table 6 jcm-09-02374-t006:** MIS across country of birth post-IMMRP (n = 14,546) and 12-month follow-up (n = 8785). ANOVA and post-hoc tests are to the far right.

Country of Birth	Sweden			Other Nordic			Europe			Outside Europe			ANOVA	
Variables	*N*	Mean	SD	*N*	Mean	SD	*N*	Mean	SD	*N*	Mean	SD	*p*	post hoc
MIS post-IMMRP	11,885	0.01	2.58	406	0.07	2.40	743	−0.20	2.63	1512	−0.12	2.67	0.053	NA
MIS at 12-m FU	7284	0.03	2.78	251	0.15	2.77	435	−0.41	2.86	815	−0.27	2.90	<0.001	SE NE Europe (*p* = 0.009) and outside Europe (*p* = 0.025)

FU = follow-up; SE = Sweden; NE = not equal.

**Table 7 jcm-09-02374-t007:** Input variables (in bold type) together with cluster sizes for the five identified clusters. In the lower part of the table are the proportions participating in IMMRP, age, number of days with no work, and pain duration. Chi^2^ and ANOVA are to the far right. For age, number of days with no work, and pain duration are given effect sizes (ES, Hedges’g) for cluster 1 vs. cluster 5.

Variable	Cluster 1	Cluster 2	Cluster 3	Cluster 4	Cluster 5	Statistics
*N* (%)	6356 (16.4%)	5091 (13.1%)	13,034 (33.6%)	8867 (22.9%)	5442 (13.4%)	
Women (%)	100	100	100	0	65.2	Chi^2^ = 32628.8, df = 4, *p* > 0.001
Born outside Europe (%)	0	0	0	0	100	Chi^2^ = 38790.0, df = 4, *p* > 0.001
Education level (%):						Chi^2^ = 49410.0, df = 8 *p* > 0.001
Elementary school	0	100	0	24.1	28.4	
Upper Secondary school	0	0	100	58.8	46.4	
University	100	0	0	17.0	25.2	
Participated in MMRP (%)	42.5	35.2	42.3	33.4	28.2	Chi^2^ = 472.0, df = 4, *p* > 0.001
Age (years; mean ± SD)	44.07 ± 10.5	43.74 ± 13.11	41.01 ± 11.31	44.37 ± 11.62	44.42 ± 9.39	ANOVA: *p* < 0.001; ES = 0.03
Days with no work (mean ± SD)	1076 ± 1944	1852 ± 2602	1322 ± 2543	1301 ± 2361	1231 ± 2146	ANOVA: *p* < 0.001; ES = 0.08
Pain duration (days; mean ± SD)	3221 ± 3505	3494 ± 3537	3104 ± 3176	3105 ± 3453	2571 ± 2824	ANOVA: *p* < 0.001; ES = 0.19

Chi^2^ = Chi square; df = degrees of freedom; ES = effect size.

**Table 8 jcm-09-02374-t008:** Baseline data for the five identified clusters based on sex, education level, and country of birth as input variables. Statistics are furthest to the right—i.e., group comparison (ANOVA) and effect size (ES; Hedges’g).

Clusters	Cluster 1		Cluster 2		Cluster 3		Cluster 4		Cluster 5		ANOVA		Cl1 vs. Cl5
Variables	Mean	SD	Mean	SD	Mean	SD	Mean	SD	Mean	SD	*p*-value	post hoc	ES
NRS-7d	6.49	1.79	7.32	1.65	7.05	1.67	6.67	1.84	7.95	1.62	<0.001	all different	0.85 ***
HAD-A	8.17	4.67	9.20	4.88	8.85	4.87	8.51	4.78	12.71	4.79	<0.001	all different	0.96 ***
HAD-D	7.85	4.43	8.66	4.59	8.36	4.61	8.62	4.83	10.82	4.72	<0.001	cl2 = cl4; other different	0.65 **
MPI-Pain sever	4.16	0.96	4.65	0.88	4.51	0.89	4.29	1.00	4.94	0.86	<0.001	all different	0.85 ***
MPI-Pain interfere	4.19	1.14	4.49	1.04	4.41	1.06	4.35	1.12	4.75	0.94	<0.001	all different	0.53 **
MPI-control	2.81	1.10	2.58	1.19	2.68	1.14	2.74	1.19	2.26	1.23	<0.001	all different	0.47 *
MPI-distress	3.24	1.29	3.51	1.34	3.45	1.32	3.35	1.34	4.21	1.20	<0.001	cl2 = cl3; other different	0.78 **
MPI-SOCSupp	3.90	1.38	4.18	1.42	4.13	1.39	4.27	1.36	4.41	1.40	<0.001	cl2 = cl3; other different	0.37 *
MPI-punish	1.66	1.39	1.70	1.39	1.69	1.40	1.89	1.33	2.02	1.53	<0.001	cl1 = cl2 and 3, cl2 = cl3; other different	0.25 *
MPI-protect	2.71	1.32	3.04	1.49	2.99	1.46	2.82	1.36	3.80	1.49	<0.001	cl2 = cl3; other different	0.78 **
MPI-distract	2.38	1.17	2.52	1.24	2.52	1.21	2.49	1.18	3.07	1.40	<0.001	cl2 = cl 3 and cl4, cl3 = cl4, other different	0.54 **
MPI-GAI	2.48	0.81	2.35	0.86	2.45	0.84	2.32	0.91	1.97	1.09	<0.001	cl1 = cl3, cl2 = cl4; other different	0.54 **
EQ-5D-index	0.30	0.32	0.23	0.30	0.25	0.31	0.24	0.32	0.12	0.30	<0.001	cl2 = cl4, cl3 = cl4; other different	0.58 **
EQ-VAS	42.87	19.49	39.03	19.92	40.75	19.64	41.15	20.73	36.23	21.47	<0.001	cl3 = cl4; other different	0.33 *
sf36-pf	55.37	21.74	46.64	21.45	51.19	21.30	54.11	23.36	42.55	22.12	<0.001	all different	0.58 **
sf36-rp	13.84	25.83	12.73	25.35	13.56	25.51	14.45	27.00	11.37	24.67	<0.001	cl5 NE cl1, cl3, cl4; cl2 NE cl4	0.10
sf36-bp	26.53	15.49	22.03	14.69	23.34	14.39	25.28	15.65	17.86	14.66	<0.001	all different	0.57 **
sf36-gh	43.41	21.29	37.38	20.47	40.53	20.79	43.31	20.38	29.85	19.05	<0.001	cl1 = cl4, other different	0.67 **
sf36-vt	24.58	19.31	21.88	18.36	22.67	18.50	27.40	20.32	23.25	18.98	<0.001	cl2 = cl3, cl3 = cl5; other different	0.07
sf36-sf	46.83	26.05	46.18	26.03	47.31	25.88	48.99	27.23	38.28	24.76	<0.001	cl1 = cl2 and cl3, cl2 = cl3; other different	0.34 *
sf36-re	50.43	43.41	40.76	43.39	44.45	43.44	43.40	43.48	22.95	36.46	<0.001	cl3 = cl4; other different	0.68 **
sf36-mh	58.46	20.67	53.94	22.08	55.81	22.01	55.31	22.91	40.57	22.25	<0.001	cl3 = cl4; other different	0.84 ***

Cl = cluster; Cluster 1: Women born in Europe with University education, n: 5749–6306; Custer 2: Women born in Europe with elementary school, n: 4503–5022; cluster 3: Women born in Europe with Upper Secondary school, n: 11,890–12,888; cluster 4: European men with different education levels, n: 7657–8718; cluster 5: Women and men born outside Europe with different education levels, n: 4 482–5 304; NE = not equal;; * = small effect ES, ** = medium effect ES, *** large effect ES.

**Table 9 jcm-09-02374-t009:** Outcomes of IMMRP according to MIS and change variables (change-pain and change-life) in the five clusters post-IMMRP and at 12-month follow-up. Statistics are furthest to the right, i.e., group comparison (ANOVA and Chi^2^ test). For MIS is also calculated pairwise effect sizes (Hedges’g) for certain clusters.

Clusters	Cl1			Cl2			Cl3			Cl4			Cl5			ANOVA		ES	ES
Variables	*N*	Mean	SD	*N*	Mean	SD	*N*	Mean	SD	*N*	Mean	SD	*N*	Mean	SD	*p*-value	post hoc	cl1 vs. cl2	cl1 vs. cl5
MIS																			
MIS post-IMMRP	2695	0.18	2.63	1780	−0.16	2.56	5486	0.00	2.56	2947	−0.07	2.55	1501	−0.12	2.66	<0.001	cl 1 NE cl2-5	0.13	0.11
MIS 12-month FU	1656	0.34	2.86	1098	−0.29	2.61	3383	0.06	2.75	1745	−0.19	2.84	810	−0.26	2.91	<0.001	cl2, cl4, cl5 = cl2, cl 4, cl5	0.23 *	0.21 *
		%			%			%			%			%		*p*-value	Chi^2^; df		
Pain-change (% diminished pain)																			
Post IMMRP		61.9			53.0			55.3			57.2			55.9		<0.001	49.5; 8		
12-month FU		63.4			48.2			56.3			58.0			50.7		<0.001	113.2; 8		
Life-change (% improved)																			
Post-IMMRP		88.6			81.0			85.6			81.9			77.3		<0.001	118.9; 8		
12-month FU		83.5			71.6			79.7			72.2			65.7		<0.001	169.3; 8		

Cl = cluster; FU = follow-up; ES = effect size; * = small effect ES.

## Data Availability

The datasets generated and/or analyzed in this study are not publicly available as the Ethical Review Board has not approved the public availability of these data.

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
