# Peer review of "Influences of Sex, Education, and Country of Birth on Clinical Presentations and Overall Outcomes of Interdisciplinary Pain Rehabilitation in Chronic Pain Patients: A Cohort Study from the Swedish Quality Registry for Pain Rehabilitation (SQRP)"

_jcm, 2020, doi:10.3390/jcm9082374_

Round 1

Reviewer 1 Report

This is a register based study, with a large sample size. It is well written and holds a methodologically high standard. Evaluating the  response to treatment and identifying subgroups of patients is of major importance. The limitations in this type of  studies are the  lack of knowledge/control regarding the intervention delivered to the different  patients and subgroups of patients. Does the register contain information regarding the delivered treatments? This may be  over more importance than every detail of the 22 outcome measurements reported.  At least a bit more thorough discussion regarding this issue. In particular one may wonder if treatment delivered is systematically different across patient cultural background, or if compliance varies.

Author Response

Reviewer 1 – point-to-point reply

This is a register based study, with a large sample size. It is well written and holds a methodologically high standard. Evaluating the  response to treatment and identifying subgroups of patients is of major importance. The limitations in this type of  studies are the  lack of knowledge/control regarding the intervention delivered to the different  patients and subgroups of patients. Does the register contain information regarding the delivered treatments? This may be  over more importance than every detail of the 22 outcome measurements reported.  At least a bit more thorough discussion regarding this issue. In particular one may wonder if treatment delivered is systematically different across patient cultural background, or if compliance varies.

Our answer and revisions: Thank you for the positive comments about our study.
Due to this comment we have added the following sentences in the discussion (section 4.6 Strengths and limitations): Although all specialist clinics' IMMRPs can be included in the general description of IMMRP (see introduction), there may be heterogeneity regarding e.g. scope, and intensity of the different components of IMMRP as well as different competence of the therapists. Furthermore, the team's internal interaction, interaction with the patient and interaction with other relevant actors e.g. the employers and the Social Insurance Agency, can differ. At present, no detailed information is available within the registry that captures these aspects which is a limitation. (Lines 598-604)

Reviewer 2 Report

Thank you for the opportunity to review the article Influences of Gender, Education, and Country of Birth on Clinical Presentations and Overall Outcomes of Interdisciplinary Pain Rehabilitation in Chronic Pain Patients: A Cohort Study From the Swedish Quality Registry for Pain Rehabilitation (SQRP) for the Journal of Clinical Medicine. This article fills an important gap in understanding who is most likely to benefit from participating in IMMRPs, in their current form, in Sweden and for whom programs need to be adapted in order to more successfully support patients from diverse backgrounds. The article highlights that IMMRPs do not help patients from various sociodemographic backgrounds equally and improvements are needed.

Below is a list of major and minor concerns that we feel the authors should consider addressing:

Major Concerns

  1. The authors should provide additional background information on who has access to the specialist clinics and IMMRPs in Sweden.
    1. For example, are visitors from other countries able to access care in Sweden or are only Swedish residents?
    2. What is the process of enrollment in an IMMRP? Do you have to first attend a primary care appointment, receive a referral to a specialist clinic and then receive a recommendation from the specialist in order to enroll in an IMMRP?
    3. Is the group that participates in IMMRP representative of the larger sample that are in the registry? Are there differences in demographic features for who participates in IMMRPs compared to the larger registry sample? Any differences should be accounted for when describing differences across demographic variables at baseline.
    4. Might language barriers contribute to experience? Are all patients expected to communicate in Swedish or can care be provided in other languages? This would help clarify if there are additional access barriers contributing to differences clinical presentations across participants.
  2. Towards the end of the introduction, include more of a transition between discussion of sociodemographic factors: gender, education, country of birth. Additionally, there is a much greater focus on gender in the introduction, but the paper places equal weight between the three sociodemographic factors, thus the introduction should reflect this and more attention should be given to potential roles of education and country of birth.
  3. The choice for the three primary outcomes needs more support. The authors describe a Multivariate Improvement Score that encompasses 22 outcomes, but then present these 22 outcomes separately. Additionally, why are two outcomes the patient-reported retrospective estimates on improvement? This subjective variable seems less desirable when compared to the pain intensity, pain interference, functioning, and quality of life measures included at baseline and follow-up timepoints. These measures are continuous variables and would offer more variability than the trichotomized estimated change variables.

Minor Concerns

Introduction

  1. Line 44- 45, awkward phrasing
  2. Line 48-49, add and: Acceptance and Commitment Therapy
  3. Line 84, special should be specialty
  4. Line 89, consider discussing “change in life situation” earlier in the introduction as there is no context for this variable.

Experimental Section

  1. Line 97, is Patient Reported Outcome Measures (PROM) the PROMIS measures (Patient Reported Outcome Measure Information System)? If so, use full name and abbreviation throughout paper.
  2. Line 112-114, please cite the literature that states that patients referred to specialist care often have comorbidities such as depression, anxiety, and kinesiophobia.
  3. Line 114-115, the way the inclusion criteria is described is confusing. It is not clear why strict criteria were not available because of the registry population. Consider simply, “There were no strict inclusion criteria.”
  4. Line 121, it is not clear what is meant by the example “red flags.” Please explain or consider providing an example of a condition with another treatment option.

Results

  1. Consider adding indicators for significant p-values or indicators for the size of effects (e.g., *) throughout tables.
  1. Consider using more recognizable variable names for the 22 outcomes variables, especially the sf36 variables or including note under Table 1. With reference for what each variable abbreviation stands for.

Discussion

  1. Line 455-458, not clear what the goal of this sentence is. It does not clearly explain why non-Europeans have worse clinical presentations.
  2. In thinking about results found regarding country of birth, it would be helpful to know if there was an interaction effect between education and country of birth. A potential explanation of results may include discussion of SES and education related disadvantages associated with being an immigrant.
  3. Lines 468-469, consider using language such as bias, culturally insensitive programming, discriminatory attitudes, or not culturally competent providers
  4. Line 483, add “education” after “elementary school”
  5. Line 485-486, reiterate the gender difference concisely (e.g., more females) rather than refer to previous report
  6. Line 490, what is meant by gendered norms and how would they relate to the attitudes of the team? Are you referring to bias here?
  7. Line 491- 494, this sentence is an incomplete thought and needs to be expanded to clarify what is being stated.
  8. Line 494, consider “providers” rather than “caregivers”, which might suggest family members.
  9. Lines 501-503 the authors begin to provide explanation for results of cluster analyses, but additional discussion of explanation for the findings would be helpful.
  10. Finish main discussion section with a statement describing how this information can be used to help design IMMRPs or what more we need to know in order to use this information about differences by sociodemographic factors- how are each factor contributing to worse outcomes?

Reviewer 3 Report

1. Throughout the manuscript, you are using the term gender instead of sex. Please correct your title and manuscript accordingly. 2. What was the reason for having a majority of your patients being women (76.3%)? It is really hard to talk about influences of sex when you have more than 2/3 of your population being women. 3. Adding 5 clusters to your analysis makes your results more confusing, especially the way you combined different populations in Cluster #5.  4. Even though you briefly mentioned the difference between statistical and clinical significance, you should emphasize the possibility of getting false positive results when you have such a large patient population more. 

Round 2

Reviewer 2 Report

Thank you for this quick revision and for including additional explanations where necessary. The authors have done an admirable job addressing the reviewers’ suggestions, particularly in expanding the context for the study in the introduction and the discussion. I have a few remaining concerns to be considered.

  1. The authors should be commended for addressing an important set of research questions in a large database of chronic pain patients. However, the vast amount of data makes it hard to understand what the primary aim of the paper is. There is an abundance of data on “baseline clinical presentation”, and these have the strongest effect sizes. There are also numerous ways of looking at the outcomes data (MIS, change in life, change in time, at both post IMMRP and 12-months); however, the presentation and interpretation of this data is confusing for a few reasons:
    1. We don’t know if there are systematic differences in who is attending these programs. Just as you did for the cluster analysis (Table 7), it would be helpful to know what percentage of males vs females, how many from each education level, and from each country of birth attended IMMRPs.
    2. The outcomes data you present for each demographic factor is different. Please present results similarly across demographic factors. Table 9 does a nice job of presenting both overall MIS, pain-change and life-change outcomes. Table 2 presents change-life and change-pain clearly, but MIS data is missing. Either presentation of change outcomes is acceptable, but be consistent. .
    3. Related to item b., despite saying that you will analyze the 22 specific MIS outcomes in more depth, you only look at differences in both baseline levels and outcomes for sex. It is unclear why you did not do this for education level and country of birth groups. Please be consistent across demographic factors.

One possible way to streamline  presentation is to only include tables with the 22 specific MIS outcomes (including baseline data) in the supplementary tables, and only include MIS, change - pain and life , and demographic data in the tables in the text.

Another possible suggestion is to organize the discussion so that you focus first on demographic differences at baseline first (sex, education, and country of birth), and then to discuss the outcome data together in light of the limitations (different programs, differences and limitations in who attends IMMRPs).

  1. Although described in the previous study, the MIS needs more explanation regarding how to interpret the primary outcomes in this study. It would be helpful for readers to know that positive scores mean improvement per Ringqvist 2019. Additionally, the discussion of the cluster analysis from the previous publication (page 5, lines 225-231) is confusing given that the current study includes a different cluster analysis. Please clarify.
  2. I believe that reviewer 3 was addressing the fact that most health providers assess for biological sex first and may not ask for gender identity. Thus, they may have been suggesting that the manuscript use the word “sex” in place of “gender” if that in fact more accurately describes the data that were collected.
  3. Line 44- 45, still slightly awkward phrasing, consider…and are combined with the individual goals of the patient.
  4. Line 48-49, I was looking for you to add the word “and” as it is called Acceptance AND Commitment Therapy
  5. Line 142. Thank you for the clarification in text. Consider removing “red flags” so the line reads “specific pain conditions with other treatment options available (i.e., instance saddle anaesthesia or history of carcinoma)”
  6. Consider using a different amount of stars to indicate different sizes of effects in your tables * = small effect, ** = medium effect, *** large effect
  7. Thank you for adding “elementary school education” in line 483. I see that “education” is needed after elementary school In line 578, as well
  8. In your discussion of outcomes, you overstate sex differences, saying that there were improvements across the 22 specific MIS outcomes (line 490), when in fact only a handful of differences in outcomes between men and women emerged, with very small effect sizes. Please edit the line accordingly.
  9. Thank you for revising the sentences lines 522-528. However, the last two sentences now appear to be incomplete. Please revise for clarity.
  10. Lines 582-587 Thank you for providing more explanation regarding the cluster analyses. I was looking for you to provide an explanation as to why European women with university education had the best results versus why European women with elementary school education and non-European patients had the worst outcomes. The last sentence of this addition begins to do this. Consider editing above that sentence. Additional suggested edits for this addition include choosing a different phrase than “cluster belonging”- its meaning is unclear. Are you trying to say something like it will be important to understand the additive effect of each sociodemographic factor within the context of the other factors within the cluster”?  I would consider combining the idea of giving important clues as to how to improve the outcomes of IMMRPs with the sentence before this statement.
  11. The clinical implications section is a very nice addition.

Reviewer 3 Report

Thank you for addressing majority of my comments.

Please correct "gender" into "sex" because you are talking about two different biological sexes. Gender is not anymore binary variable and in order to avoid confusions, please make suggested chameg 
